# Serum levels of NLRC4 and MCP-2/CCL8 in patients with active Crohn's disease

**Kader Irak**[1]*, **Mehmet Bayram**[1], **Sami Cifci**[1], **Gulsen Sener**[2]

**1** Department of Gastroenterology, Basaksehir Cam and Sakura City Hospital, Istanbul, Turkey,
**2** Department of Biochemistry, Basaksehir Cam and Sakura City Hospital, Istanbul, Turkey

* drkaderirak@hotmail.com

**Data Availability Statement:** All relevant data are within the paper and its Supporting information files.

**Funding:** The authors received no specific funding for this work.

## Abstract

Crohn's disease (CD) is characterized by malfunction of immune-regulatory mechanisms with disturbed intestinal mucosal homeostasis and increased activation of mucosal immune cells, leading to abnormal secretion of numerous pro- and anti-inflammatory mediators. MCP2/CCL8 is produced by intestinal epithelial cells and macrophages, and is a critical regulator of mucosal inflammation. NLRC4 is expressed in phagocytes and intestinal epithelial cells and is involved in intestinal homeostasis and host defense. However, no study to date has assessed the circulating levels of NLRC4 and MCP2/CCL8 in patients with CD. The study was aimed to investigate the serum levels of MCP2/CCL8 and NLRC4 in patients with active CD. Sixty-nine patients with active CD and 60 healthy participants were included in the study. Serum levels of NLRC4 and MCP2/CCL8 were determined using an enzyme-linked immunosorbent assay. The median serum NLRC4 levels were lower in the patient group than in the controls (71.02 (range, 46.59–85.51) pg/mL vs. 99.43 (range 83.52–137.79) pg/mL) ($P < 0.001$). The median serum levels of MCP2/CCL8 were decreased in patients with CD (28.68 (range, 20.16–46.0) pg/mL) compared with the controls (59.96 (range, 40.22–105.59) pg/mL) ($P < 0.001$). Cut-off points of NLRC4 (<81 pg/mL) and MCP2/CCL8 (<40 pg/mL) showed high sensitivity and specificity for identifying active CD. In conclusion, this is the first study to examine circulating levels of MCP2/CCL8 and NLRC4 in patients with active CD. Our results suggest that serum NLRC4 and MCP2/CCL8 levels may be involved in the pathogenesis of CD and may have a protective effect on intestinal homeostasis and inflammation. Serum levels of MCP2/CCL8 and NLRC4 could be used as a diagnostic tool and therapeutic target for CD.

## Introduction

Crohn's disease (CD) is a chronic inflammatory condition characterized by transmural, remitting and relapsing, focal inflammation involving different sites of the gastrointestinal tract, from the mouth to the anus [1]. The epidemiology of the disorder, traditionally regarded as a condition of developed countries, is changing around the world at the beginning of the 21st century with the rapid increase in the incidence of newly industrialized countries in Asia, South America, and Africa [2]. Rapid changes in CD epidemiology have led to a global

**Competing interests:** The authors have declared that no competing interests exist.

challenge for disease diagnosis, management, and prevention, as well as creating a significant socioeconomic burden for healthcare systems. Although the risk of development and progression of CD might be affected by environmental and nutritional factors, immunologic characteristics, impaired intestinal barrier function, and gut microbiota dysbiosis in genetically susceptible individuals, the exact etiopathogenesis remains unknown [3]. Abnormal expressions of pro- and anti-inflammatory molecules including cytokines, chemokines, inflammasomes, miRNAs, and neuropeptides in both innate and adaptive immune responses are the central driver of CD, suggesting that these molecules may be used as potential diagnostic and therapeutic targets for future genetic and immunologic research [4].

Chemokines are a large family of small proteins that play a significant role in mucosal homeostasis and inflammation. Chemokines are secreted by both resident mucosal cells and a wide range of infiltrating cells including T cells, macrophages, and neutrophils, which are present in CD lesions [5]. Chemokines can attract inflammatory cells into CD tissue lesions and regulate tissue-specific and immune cell selective trafficking and leukocyte retention at the site of inflammation, contributing to the tissue destructive processes in the development and progression of CD [6]. Chemokine (C-C motif) ligand 8 (CCL8), also known as monocyte chemoattractant protein 2 (MCP2), is a small cytokine belonging to the CC chemokine family, which is involved in the inflammatory response by attracting and activating many different immune cells including monocytes, T cells, natural killer cells, eosinophils, and basophils. Experimental studies have shown that MCP2/CCL8 is produced by intestinal epithelial cells and macrophages and is a critical regulator of mucosal inflammation [7]. However, circulating levels of MCP2/CCL8 have not been investigated in patients with CD.

Inflammasomes are cytosolic multiprotein complexes that detect and eliminate both pathogens and damage-associated molecular patterns in innate immunity [8]. Inflammasome activation can lead to pyroptosis, a unique form of cell death with characteristics of classic apoptosis (caspase activation) and necrosis (membrane permeabilization) [9]. The impairment of inflammasome activation may contribute to the pathogenesis of various conditions including cancer, autoimmune disorders, metabolic, and neurodegenerative diseases [10]. Recent studies have shown that inflammasome activity is essential for intestinal homeostasis and discrimination of harmful pathogens and commensals [11]. Inflammasomes consist of nucleotide-binding and oligomerization domain-like receptor family (NLR) proteins such as NLRP1, NLRP3, NLRP6, and NLR-caspase activation and recruitment domain-containing protein 4 (NLRC4). NLRC4, also known as interleukin (IL)-converting enzyme protease-activating factor, is expressed in phagocytes and intestinal epithelial cells and is involved in intestinal homeostasis and host defense, especially against enteric pathogens [12]. It was reported that the mutation of NLRC4 could cause a syndrome of enterocolitis and autoinflammation [13]. However, no study to date has assessed the circulating levels of NLRC4 in patients with CD.

The aim of the study was to investigate the serum levels of MCP2/CCL8 and NLRC4 in patients diagnosed as having active CD.

## Material and methods

This study was conducted between July 2020 and January 2021 in the Departments of Internal Medicine and Gastroenterology of Kanuni Sultan Suleyman Training and Research Hospital, Istanbul, Turkey. Sixty-nine patients with active CD and 60 healthy control participants were included in the study. CD was diagnosed in accordance with the criteria reported by the European Crohn´s and Colitis Organization (ECCO), based on anamnesis, physical examination, magnetic resonance imaging, as well as gastrointestinal tract endoscopic examinations

including gastroscopy and colonoscopy with histopathologic evaluations [14]. Disease activity was determined using the CD Activity Index score (CDAI), and patients with a score of more than 200 points for a minimum of 2 weeks before randomization were defined as having active CD [15]. Participants with acute or chronic infections, malignancy, chronic inflammatory conditions, pregnant patients and those within 1 year postpartum, a history of concomitant autoimmune diseases, and also those in remission of CD were excluded from the study. The control group comprised healthy individuals with no known gastrointestinal disorders or history of chronic or autoimmune disease, as well as medication use. Clinical and demographic characteristics including age, sex, current medication, the sites of gastrointestinal involvement, Column L, Column M, and CD type were obtained from patients' records. All research procedures were evaluated and accepted by the Research Ethics Committee of Kanuni Sultan Suleyman Training and Research Hospital and were conducted in agreement with the ethical standards specified in the Declaration of Helsinki. Written and verbal informed consent was obtained from all participants before they participated in the study.

Blood samples were drawn from the antecubital vein after a 12-hour fast and were centrifuged at 3000 rpm for 10 min to separate the serum. Serum samples for NLRC4 and MCP2/CCL8 were stored at -40°C until the analyses were completed. Complete blood count and serum C-reactive (CRP) levels were measured on the sampling day. Complete blood count including white blood cell count (WBC), red blood cell count (RBC), red blood cell distribution width (RDW), hemoglobin values, leukocytes subtypes, platelet counts, and mean platelet volume (MPV) were measured using a Mindray BC-6800 autoanalyzer (Mindray Electronics Co, Ltd, Shenzhen, China). The neutrophil-to-lymphocyte ratio (NLR) was calculated as a simple ratio of the absolute counts of neutrophils divided by absolute lymphocyte counts, and the platelet-to-lymphocyte ratio (PLR) were determined by dividing the platelet count by the lymphocyte count. Serum CRP levels were determined using an immunoturbidimetric method with an Olympus AU 400 biochemical autoanalyzer (Olympus Optical, Tokyo, Japan) and expressed as mg/L.

After thawing frozen serum samples, serum NLRC4 was determined using an enzyme-linked immunosorbent assay (ELISA) kit according to the manufacturer's instructions (Sino-Geneclon Biotech, HangZhou, China). The detection range of the NLRC4 assay was 7.5–1200 pg/mL, and the analytical sensitivity was 3.9 pg/mL. Serum MCP2/CCL8 levels were measured using ELISA kits based on the instruction manual (SinoGeneclon Biotech, HangZhou, China). The analytical sensitivity of the MCP2/CCL8 was 0.8 pg/mL, and the detection range was 3.3–200 pg/mL. The intra-assay coefficient of variation of NLRC4 and MCP2/CCL8 was <8%, and the inter-assay coefficient of variation was <10%.

## Statistical analyses

All analyses were performed on the SPSS v21 software (SPSS Inc., Chicago, IL, USA). The Kolmogorov-Smirnov test was used to check normality of distribution. Data are given as mean ± standard deviation (SD) or median (1st quartile—3rd quartile) for continuous variables according to the normality of distribution and as frequency (percentage) for categorical variables. Normally distributed variables were analyzed using the independent samples t-test. Non-normally distributed variables were analyzed using the Mann-Whitney U test. Categorical variables were analyzed using Chi-square tests or Fisher's exact test. Spearman correlation coefficients were calculated for the assessment of relationships between continuous variables. The diagnostic performances of the variables were evaluated using receiver operating characteristic (ROC) curve analysis. P-values <0.05 were accepted as statistically significant.

**Table 1. Demographic and biochemical characteristics of participants.**

| | Control | Crohn's disease | p value |
|---|---|---|---|
| | (n = 60) | (n = 69) | |
| Age | 29 (26–35) | 34 (29–42) | **0.004** |
| Sex | | | |
| Female | 29 (48.33%) | 16 (23.19%) | **0.005** |
| Male | 31 (51.67%) | 53 (76.81%) | |
| CRP | 0.84 (0.44–1.56) | 28.00 (12.00–79.00) | <**0.001** |
| Hemoglobin | 14.12 ± 1.54 | 12.14 ± 2.36 | <**0.001** |
| RBC (x10$^6$) | 4.90 ± 0.50 | 4.71 ± 0.72 | 0.087 |
| RDW | 12.6 (12.2–13.2) | 14.7 (13.5–16.8) | <**0.001** |
| MPV | 10.14 ± 0.80 | 9.51 ± 0.92 | <**0.001** |
| Platelet (x10$^3$) | 255.57 ± 48.86 | 374.62 ± 155.31 | <**0.001** |
| WBC (x10$^3$) | 6.69 (5.72–7.90) | 9.00 (7.50–12.20) | <**0.001** |
| Neutrophils (x10$^3$) | 3.56 (3.14–4.51) | 6.20 (4.70–8.30) | <**0.001** |
| Lymphocyte (x10$^3$) | 2.45 ± 0.76 | 1.89 ± 0.72 | <**0.001** |
| NLR | 1.57 (1.23–1.85) | 3.43 (2.45–5.00) | <**0.001** |
| PLR | 112.15 (82.20–136.62) | 193.13 (137.50–265.33) | <**0.001** |
| NLRC4 | 99.43 (83.52–137.79) | 71.02 (46.59–85.51) | <**0.001** |
| MCP2/CCL8 | 59.96 (40.22–105.59) | 28.68 (20.16–46.00) | <**0.001** |

CRP: C-reactive protein, RBC: Red blood cell, RDW: Red cell distribution width, MPV: Mean platelet volume, WBC: White blood cell, NLR: Neutrophil to Lymphocyte ratio, PLR: Platelet to lymphocyte ratio, NLRC4: Nucleotide-binding and oligomerization domain-like receptor family CARD domain-containing protein 4, MCP2/CCL8: Monocyte chemotactic protein 2/ chemokine ligand 8. Data are given as mean ± standard deviation or median (1st quartile—3rd quartile) for continuous variables according to the normality of distribution and as frequency (percentage) for categorical variables.

## Results

A total of 69 patients with CD and 60 healthy participants were included in the study. The median age of the patients with CD was 34 (range, 29–42) years, and for the healthy individuals, it was 29 (range, 26–35) years. Fifty-three (76.81%) patients with CD and 31 (51.67%) controls were male. A statistically significant difference was found regarding age and sex between the two groups ($P < 0.004$ and $0.005$, respectively) (Table 1).

NLRC4 and MCP2/CCL8 levels were found to be negatively correlated with age, CRP, RDW, NLR, PLR, and positively correlated with platelet count, WBC, neutrophil count, and lymphocyte count ($p < 0.05$) (Table 2).

The sites of gastrointestinal involvement of the patients were as follows: ileocolonic (n = 22, 31.88%), colonic (n = 21, 30.43%), ileocecal (n = 13, 18.84%), and ileal (n = 13, 18.84%) (Table 2). Thirty-four patients presented with inflammatory CD, 17 had fistulizing CD, and 16 patients had stricture formation. Thirty-one patients received treatment for CD (Table 3).

The median serum NLRC4 levels were found to be lower in patients with CD than in the healthy participants (71.02 (range, 46.59–85.51) pg/mL vs. 99.43 (range, 83.52–137.79) pg/mL) ($P < 0.001$). The median serum levels of MCP2/CCL8 were significantly lower in the patient group (28.68 (range, 20.16–46.0) pg/mL) compared with the control group (59.96 (range, 40.22–105.59) pg/mL) ($P < 0.001$). The median levels of serum CRP, RDW, WBC, neutrophils, NLR, and PLR were significantly higher in patients with CD than in the controls (all, $P < 0.005$). The mean hemoglobin and MPV levels were lower in patients with CD (all, $P < 0.005$). In addition, platelet counts were 374.62 ± 155.31 x10$^3$ /μL in the patient group and 255.57 ± 48.86 x10$^3$ /μL in the control group ($P < 0.001$). The mean RBC values were found similar

**Table 2. Correlation between NLRC4 and MCP2/CCL8 levels and other parameters.**

| | NLRC4 | | MCP2/CCL8 | |
| --- | --- | --- | --- | --- |
| | r | p | r | p |
| NLRC4 | - | - | 0.755 | **<0.001** |
| Age | -0.208 | **0.018** | -0.275 | **0.002** |
| CRP | -0.450 | **<0.001** | -0.332 | **<0.001** |
| Hemoglobin | 0.250 | **0.004** | 0.053 | 0.550 |
| RBC | 0.108 | 0.224 | -0.031 | 0.729 |
| RDW | -0.379 | **<0.001** | -0.188 | **0.033** |
| MPV | 0.073 | 0.413 | 0.069 | 0.434 |
| Platelet | 0.425 | **<0.001** | 0.421 | **<0.001** |
| WBC | 0.376 | **<0.001** | 0.384 | **<0.001** |
| Neutrophils | 0.373 | **<0.001** | 0.391 | **<0.001** |
| Lymphocyte | 0.398 | **<0.001** | 0.361 | **<0.001** |
| NLR | -0.309 | **<0.001** | -0.205 | **0.020** |
| PLR | -0.270 | **0.002** | -0.194 | **0.027** |

CRP: C-reactive protein, RBC: Red blood cell, RDW: Red cell distribution width, MPV: Mean platelet volume, WBC: White blood cell, NLR: Neutrophil to Lymphocyte ratio, PLR: Platelet to lymphocyte ratio, NLRC4: Nucleotide-binding and oligomerization domain-like receptor family CARD domain-containing protein 4, MCP2/CCL8: Monocyte chemotactic protein 2/ chemokine ligand 8.

between the two groups ($P < 0.087$). The participants' biochemical characteristics are presented in Table 1.

Receiver operating characteristics (ROC) curve analysis was performed to determine the diagnostic accuracies of NLRC4 and MCP2/CCL8 for active CD (Table 4 **and** Fig 1). ROC curve analysis showed that <81 pg/mL NLRC4 as the cut-off point for the prediction of

**Table 3. Patients' characteristics of Crohn's disease.**

| Involvement | |
| --- | --- |
| Ileal | 13 (18.84%) |
| Ileocaecal | 13 (18.84%) |
| Ileocolonic | 22 (31.88%) |
| Colonic | 21 (30.43%) |
| Column L | |
| 1 | 5 (7.25%) |
| 2 | 30 (43.48%) |
| 3 | 29 (42.03%) |
| 4 | 5 (7.25%) |
| Column M | 9 (7–11) |
| Type | |
| Inflammatory | 34 (49.28%) |
| Stricture Formation | 16 (23.19%) |
| Fistulizing | 17 (24.64%) |
| Stricture Formation + Fistulizing | 2 (2.90%) |
| Treatment | 31 (44.93%) |

Data are given as median (1st quartile—3rd quartile) for continuous variables according to normality of distribution and as frequency (percentage) for categorical variables.

**Table 4. Measurements of performance to determine active Crohn's disease.**

|  | NLRC4 | MCP2/CCL8 |
|---|---|---|
| Cut-off | <81 | <40 |
| Sensitivity | 73.53% | 71.01% |
| Specificity | 81.67% | 76.67% |
| Accuracy | 77.34% | 73.64% |
| Positive predictive value | 81.97% | 77.78% |
| Negative predictive value | 73.13% | 69.70% |
| AUC (95.0% CI) | 0.777 (0.696–0.859) | 0.752 (0.667–0.836) |
| *p* value | <0.001 | <0.001 |

NLRC4: Nucleotide-binding and oligomerization domain-like receptor family CARD domain-containing protein 4, MCP2/CCL8: Monocyte chemotactic protein 2/ chemokine ligand 8, AUC: Area Under ROC Curve, CI: Confidence Intervals.

active CD had a sensitivity and specificity of 73.53% and 81.67%, respectively (area under the ROC curve (AUC): 0.777). The optimum cut-off point of <40 pg/L for MCP2/CCL8 had a sensitivity of 71.01% and specificity of 76.67% for diagnosing active CD, with an AUC of 0.752.

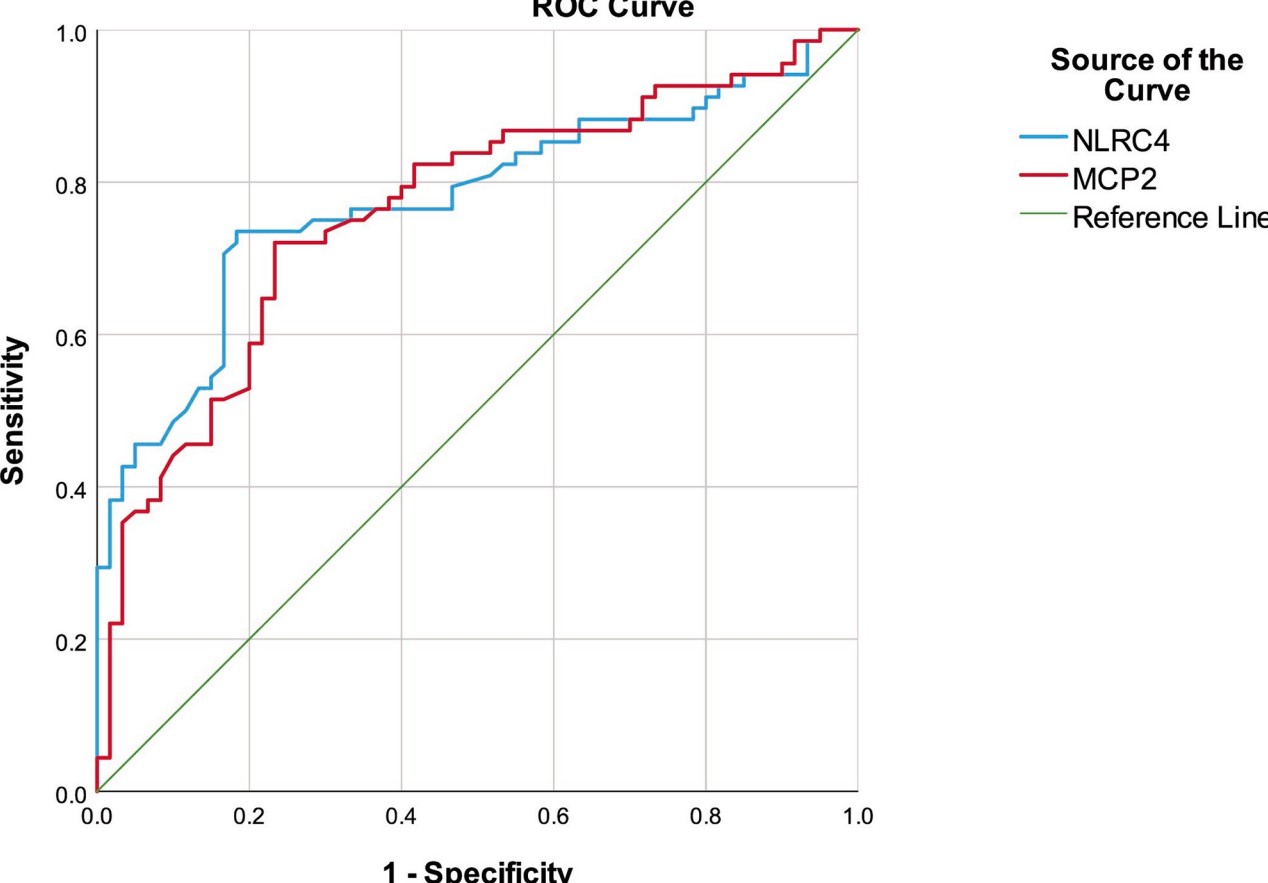

**Fig 1. ROC analysis of the NLRC4 and MCP2/CCL8 to determine active Crohn's disease.** ROC: Receiver operating characteristics, NLRC4: Nucleotide-binding and oligomerization domain-like receptor family CARD domain-containing protein 4, MCP2: Monocyte chemotactic protein 2.

## Discussion

The study is the first to evaluate serum MCP2/CCL8 and NLRC4 levels in patients with active CD. We demonstrated significantly lower serum NLRC4 levels among patients with active CD than in healthy individuals. We also demonstrated that serum MCP2/CCL8 levels were found to be significantly lower in patients with active CD compared with healthy subjects. Cut-off values of <81 pg/mL for NLRC4 and <40 pg/mL for MCP2/CCL8 showed high sensitivity and specificity in identifying active CD.

CD is characterized by malfunction of immune-regulatory mechanisms and signaling pathways with disturbed intestinal mucosal homeostasis and increased, uncontrolled activation of mucosal effector immune cells, leading to the abnormal secretion of numerous pro- and anti-inflammatory mediators that enhance the inflammatory processes [16]. However, the cellular and molecular mechanisms that trigger these processes remain largely unclear. Chemokines are inducible, immune-regulatory proteins that are required for overall systemic inflammation and homeostasis of immune functions, involving the recruitment of immune cells, as well as the adhesion and migration of leukocytes at the site of inflammation [6]. Both experimental studies and clinical investigations have reported the role of numerous chemokines in the pathogenesis of inflammatory bowel disease (IBD). However, serum levels of chemokines have been poorly investigated. Grimm et al. demonstrated increased expression and production of monocyte chemoattractant protein (MCP-1) in inflamed intestinal mucosa, suggesting the role for MCP-1 in monocyte activation and attraction to the mucosal lesions of IBD [17]. In an experimental mice model, it was shown that the absence of MCP-1 was associated with a prominent decrease in the severity of colitis and mortality [18]. In 42 patients with IBD and 10 healthy individuals, Singh et al. showed that serum levels of macrophage migration factor (MIF), CCL21, CCL23, CCL25, CXCL5, CXCL10, CXCL11, CXCL13, MCP-1 were increased in patients with IBD compared with healthy subjects [6]. Cibor et al. revealed that serum MCP-1 levels did not alter between patients with varying activity of IBD in 56 patients with ulcerative colitis (UC), 66 with CD, and 40 healthy subjects [19].

MCP2/CCL8 has been shown as an agonist of C-C chemokine receptor type 2 (CCR2), CCR3 and CCR5 and is expressed in inflamed tissues by monocyte/macrophages following paracrine stimulation from T cell lymphocytes via pro-inflammatory cytokines, or through innate mechanisms by interactions with bacterial, viral and fungal pathogens [20]. However, the cellular origin, expression mechanism, and biologic role of MCP2/CCL8 under physiologic or pathologic conditions are not fully elucidated [7]. Gong et al. found that CCL8 was a potent inhibitor of CD4/CCR-5, and mediated HIV-1 entry and replication [21]. Hori et al. demonstrated in a mouse model that serum CCL8 levels were closely correlated to graft versus host disease (GVHD) severity, suggesting that it could be used as a specific serum marker for the early and accurate diagnosis of GVHD [22]. In an experimental mouse study, Lu et al. demonstrated that colonic inflammation could induce increased expression of CCL8 and its receptor CCR5 in the neurons of the spinal column by activating the extracellular signal-regulated kinase pathway [20]. Asano et al. showed that intestinal CD169[+] macrophage-derived CCL8 functioned as a stimulant for the breakdown of barrier defense in experimentally induced colitis in mice, and anti-CCL8 neutralizing antibody was a promising target for the suppression of mucosal damage and clinical symptoms of experimentally induced colitis. [7]. Banks et al. demonstrated that the expression of MCP-2 was upregulated in patients with IBD and correlated with disease activity in an immunohistochemical study of colonic mucosal biopsy specimens [23]. We found decreased serum levels of MCP2/CCL8 in patients with active CD as compared with healthy individuals. Our results indicate that MCP2/CCL8 is involved in the pathogenesis of CD. A decrease in MCP2/CCL8 levels may affect the development and

progression of local intestinal inflammation and tissue destruction in patients with CD through cellular and molecular interactions between epithelial, immune, and inflammatory cells. Our results also suggest that MCP2/CCL8 could be used as diagnostic tool and therapeutic target for CD. It also suggested that MCP2/CCL8 may have a role in disease activity because our study group comprised patients with active CD.

Inflammasomes are multimolecular complexes that initiate innate immune responses and assemble in response to endogenous and pathogen stimuli to maintain cellular integrity and control host-microbiota interactions [8]. NLRC4 activates caspase-1, which induces the maturation and secretion of inflammatory cytokines, IL-1β, IL-18, and gasdermin D to recruit phagocytic cells, epithelial repair, angiogenesis, and the regulation of chemokines and cytokines at the site of damage [24]. NLCR4 is also involved in host intestinal defense through the discrimination of commensals and harmful intestinal pathogens [25]. Franchi et al. revealed reduced neutrophil recruitment in the Salmonella-infected NLRC4-decifient or IL-1-deficient mice compared with wild-type mice, resulting in limited intestinal inflammation and an inability to control intestinal infection [25]. Studies have reported the role of inflammasomes in the pathogenesis of IBD. Schoultz et al. demonstrated that Swedish men who had the NLRP3 polymorphism exhibited an increased risk of developing CD in 498 patients with CD and 742 controls [26]. In contrast, Lewis et al. revealed no genetic association between NLRP3 and CD in a large panel in the United Kingdom with 1298 patients with CD and 1244 healthy individuals [27]. Chen et al. showed increased serum NLRP3 levels in patients with severe UC compared with a mild/moderate group [28]. Romberg et al. demonstrated in a family that the NLRC4 mutation led to periodic fever, neonatal-onset enterocolitis, fatal or near-fatal attacks of autoinflammation, causing the production of a constitutive IL-1 family cytokine and pyroptotic cell death in macrophages [13]. Canna et al. showed that the heterozygous mutation in NLRC4 caused persistent fever and enterocolitis following parainfluenza infection in a previously healthy 6-week-old patient and was successfully treated with IL-18 inhibition [29]. We showed lower serum NLCR4 levels in patients with active CD than in healthy subjects. Our result indicates that NLRC4 may contribute to the pathogenesis of CD and may have a protective effect on intestinal homeostasis and inflammation through IL-1 and IL-18 processing in patients with CD. Our result suggests that serum levels of NLRC4 may be a useful biomarker for identifying patients with active CD.

The first limitation of our study was its relatively small sample size, which was from a single center. Secondly, we determined the disease activity by CDAI, without endoscopic evaluation. Thirdly, ELISA was used to evaluate MCP2/CCL8 and NLRC4 levels, which is not the gold standard. Finally, the difference in age, gender, and medication frequency between the case group and the control group may have affected the results.

In conclusion, this is the first study to examine circulating levels of MCP2/CCL8 and NLRC4 levels in active patients with CD. We demonstrated significantly decreased serum levels of NLRC4 and MCP2/CCL8 levels in patients with active CD compared with healthy subjects. Serum NLRC4 and MCP2/CCL8 levels are involved in the pathogenesis of CD and may have a protective effect on intestinal homeostasis and inflammation and could be used as a diagnostic tool and therapeutic target for CD.

## Supporting information

**S1 Data.**
(XLS)

## Author Contributions

**Conceptualization:** Kader Irak, Mehmet Bayram, Sami Cifci.

**Data curation:** Kader Irak, Mehmet Bayram, Sami Cifci.

**Formal analysis:** Mehmet Bayram, Sami Cifci, Gulsen Sener.

**Investigation:** Kader Irak, Mehmet Bayram, Sami Cifci.

**Methodology:** Kader Irak, Mehmet Bayram, Sami Cifci, Gulsen Sener.

**Project administration:** Kader Irak, Mehmet Bayram, Sami Cifci, Gulsen Sener.

**Software:** Kader Irak, Sami Cifci, Gulsen Sener.

**Supervision:** Kader Irak, Mehmet Bayram, Sami Cifci, Gulsen Sener.

**Validation:** Mehmet Bayram, Sami Cifci.

**Visualization:** Kader Irak, Mehmet Bayram.

**Writing – original draft:** Kader Irak, Mehmet Bayram, Sami Cifci.

**Writing – review & editing:** Kader Irak, Mehmet Bayram, Sami Cifci, Gulsen Sener.

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
