## [Decision Letter · Decision Letter 0]

2 Sep 2021

PONE-D-21-18558

Serum levels of NLRC4 and MCP-2/CCL8 in patients with active Crohn’s disease

PLOS ONE

Dear Dr. Irak,

Thank you for submitting your manuscript to PLOS ONE. After careful consideration, we feel that it has merit but does not fully meet PLOS ONE’s publication criteria as it currently stands. Therefore, we invite you to submit a revised version of the manuscript that addresses the points raised during the review process. The reviewer recommends minor revisions.

We look forward to receiving your revised manuscript.

Kind regards,

David M. Ojcius

Academic Editor

PLOS ONE

Journal Requirements:

Reviewers' comments:

Reviewer's Responses to Questions

**Comments to the Author**

1. Is the manuscript technically sound, and do the data support the conclusions?

Reviewer #1: Yes

2. Has the statistical analysis been performed appropriately and rigorously? 

Reviewer #1: Yes

3. Have the authors made all data underlying the findings in their manuscript fully available?

Reviewer #1: No

4. Is the manuscript presented in an intelligible fashion and written in standard English?

Reviewer #1: Yes

5. Review Comments to the Author

Reviewer #1: The authors studied – for the first time - the serum levels of MCP2/CCL8 and NLRC4 in active CD and found significant differences compared with the control group. This finding could be useful in clinical practice, both in diagnostic and therapy of CD.

I have some suggestions for the authors:

1. The Montreal classification of CD (location, behavior) – has no relevance in this study.

2. Please be more precious regarding the selection of the control group; as there are significant differences regarding age and sex it would be useful to check if there are any variation of studied inflammatory markers in general population related to age and sex.

3. There are some already known inflammatory markers (CRP, RDW, WBC, neutrophils, NLR, PLR, platelets) found to be different from control group but this is not relevant in this study; it would be interesting to compare these known markers with MCP2/CCL8 and NLRC4.

4. It is well known that it is no correlation between CDAI and mucosal inflammation; maybe adding calprotectin evaluation will add additional value of the study.

5. Some items in the discussion section are not supported by the results (“This supports the role of neutrophils in regulating leukocyte activation and migration in the intestinal damage, inflammation in general, and also as an indicator of disease severity”).

6. The treatment may play a role in levels of MCP2/CCL8 and NLRC4. Although the patients had active CD only 31 had treatment. Please explain.

6. PLOS authors have the option to publish the peer review history of their article (what does this mean?). If published, this will include your full peer review and any attached files.

Reviewer #1: No

---

## [Author Response · Author response to Decision Letter 0]

12 Oct 2021

Reviewer #1: The authors studied – for the first time - the serum levels of MCP2/CCL8 and NLRC4 in active CD and found significant differences compared with the control group. This finding could be useful in clinical practice, both in diagnostic and therapy of CD.

I have some suggestions for the authors:

1. The Montreal classification of CD (location, behavior) – has no relevance in this study.

The data of the study is presented in an additional excel file.

2. Please be more precious regarding the selection of the control group; as there are significant differences regarding age and sex it would be useful to check if there are any variation of studied inflammatory markers in general population related to age and sex.

As far as we examined, inflammatory marker variation between different gender and age groups has not been clearly demonstrated. Even so, the difference in age and gender frequency between the case group and the control group in our study may have affected the results. We mentioned this situation in the limitations section.

3. There are some already known inflammatory markers (CRP, RDW, WBC, neutrophils, NLR, PLR, platelets) found to be different from control group but this is not relevant in this study; it would be interesting to compare these known markers with MCP2/CCL8 and NLRC4.

Correlation analyses were added as suggested

4. It is well known that it is no correlation between CDAI and mucosal inflammation; maybe adding calprotectin evaluation will add additional value of the study.

Very few of the patients included in the study have calprotectin data, which is not sufficient for analysis.

5. Some items in the discussion section are not supported by the results (“This supports the role of neutrophils in regulating leukocyte activation and migration in the intestinal damage, inflammation in general, and also as an indicator of disease severity”).

The relevant part has been removed 

6. The treatment may play a role in levels of MCP2/CCL8 and NLRC4. Although the patients had active CD only 31 had treatment. Please explain.

Although 31 patients received treatment, they applied with disease activation. Other patients were diagnosed with CD at the time of enrollment in the study and have not received any treatment yet. The possible effect of medication differences between the groups on the results is added to the limitations section.

---

## [Decision Letter · Decision Letter 1]

2 Nov 2021

Serum levels of NLRC4 and MCP-2/CCL8 in patients with active Crohn’s disease

PONE-D-21-18558R1

Dear Dr. Irak,

We’re pleased to inform you that your manuscript has been judged scientifically suitable for publication and will be formally accepted for publication once it meets all outstanding technical requirements.

Kind regards,

David M. Ojcius

Academic Editor

PLOS ONE

Additional Editor Comments (optional):

Reviewers' comments:

Reviewer's Responses to Questions

**Comments to the Author**

1. If the authors have adequately addressed your comments raised in a previous round of review and you feel that this manuscript is now acceptable for publication, you may indicate that here to bypass the “Comments to the Author” section, enter your conflict of interest statement in the “Confidential to Editor” section, and submit your "Accept" recommendation.

Reviewer #1: All comments have been addressed

2. Is the manuscript technically sound, and do the data support the conclusions?

Reviewer #1: Yes

3. Has the statistical analysis been performed appropriately and rigorously? 

Reviewer #1: Yes

4. Have the authors made all data underlying the findings in their manuscript fully available?

Reviewer #1: Yes

5. Is the manuscript presented in an intelligible fashion and written in standard English?

Reviewer #1: Yes

6. Review Comments to the Author

Reviewer #1: It is an interesting work that evaluates, for the first time, the role of serum levels of MCP2/CCL8 and NLRC4 in active CD.

The authors responded promptly to all comments in the first revision.

Although some problems remain insufficiently evaluated (value of markers used in inactive CD compared to the control group, correlation with endoscopic activity / fecal calprotectin) I believe that, through the quality of the study and the novelty of the approach, the paper can be published.

7. PLOS authors have the option to publish the peer review history of their article (what does this mean?). If published, this will include your full peer review and any attached files.

Reviewer #1: No

---

## [Editor Report · Acceptance letter]

8 Nov 2021

PONE-D-21-18558R1 

Serum levels of NLRC4 and MCP-2/CCL8 in patients with active Crohn’s disease 

Dear Dr. Irak:

I'm pleased to inform you that your manuscript has been deemed suitable for publication in PLOS ONE. Congratulations! Your manuscript is now with our production department. 

Kind regards, 

on behalf of

Dr. David M. Ojcius 

Academic Editor

PLOS ONE